# Do Anti-Biofilm Antibiotics Have a Place in the Treatment of Diabetic Foot Osteomyelitis?

**DOI:** 10.3390/antibiotics12020317

**Published:** 2023-02-03

**Authors:** Eric Senneville, Benoit Gachet, Nicolas Blondiaux, Olivier Robineau

**Affiliations:** 1Infectious Diseases Unit, Gustave Dron Hospital, F-59200 Tourcoing, France; 2French National Referent Centre for Complex Bone and Joint Infections, CRIOAC Lille-Tourcoing, F-59000 Lille, France; 3EA2694, Lille University, F-59000 Lille, France; 4Microbiology Laboratory, Gustave Dron Hospital, F-59200 Tourcoing, France

**Keywords:** diabetic foot osteomyelitis, biofilm, antimicrobial resistance, adverse effects, rifampicin, fluoroquinolones

## Abstract

The choice of antibiotic regimens for use in patients presenting with diabetic foot osteomyelitis and their duration differs according to the situation. Antibiotics play a more important role in the medical option where no infected bone has been resected, while their role is reduced but not negligible in the case of surgical options. Some studies have reported the presence of biofilm structures in bone samples taken from patients with diabetic foot osteomyelitis, which raises the question of the place of anti-biofilm antibiotic regimens in this setting. During the last two decades, clinical studies have suggested a potential benefit for anti-biofilm antibiotics, mainly rifampicin against staphylococci and fluoroquinolones against gram-negative bacilli. However, no data from randomized controlled studies have been reported so far. The present work provides a summary of the available data on the question of the place of anti-biofilm antibiotics for the treatment of diabetic foot osteomyelitis, but also the potential limitations of such treatments.

## 1. Introduction

About 20% of mild diabetic foot infections (DFIs) and up to 60% of severe cases present concomitant bone involvement [1,2]. The outcome of osteomyelitis complicating a diabetic foot ulcer (DFU) is usually worse than that of infected DFUs involving only skin and soft tissues (ST-DFIs) as diabetes foot osteomyelitis (DFO) is associated with a higher (i) length of hospital stay; (ii) total duration of antibiotic therapy; (iii) time to wound healing after admission; (iv) total duration of the wound; and (v) risk of minor and major amputation of the foot [3]. While suspicion of DFO is generally based on clinical and radiological elements, a diagnosis of certitude requires microbiological and histological criteria [4]. Bone necrosis and difficulties in achieving high local concentrations of anti-infective agents, in addition to peripheral artery disease and impaired leucocyte function generally associated with diabetes, increase the risk of poor outcomes. Biofilm-related osteomyelitis represents another cause for the poor outcome of patients treated for DFO. The presence of biofilm is one of many other barriers for antibiotics to achieve their antibacterial activity. Whether the presence of biofilms in infected bone tissues is a contraindication for the medical (i.e., with no bone resection) management of DFOs is a matter of debate. This raises the question to what extent antibiotics with anti-biofilm activity add value to the medical management of DFOs.

This present work aims to summarize the current data on the structural particularities of DFO, including biofilms, and discusses potential consequences regarding antibiotic use.

## 2. Microbiology

The microbiology of DFO is usually polymicrobial [5,6,7,8]. In almost all the reported series in Western countries, *Staphylococcus aureus* is the most common pathogen cultured from bone samples, while in warm climate countries gram-negative bacilli dominate, especially *Pseudomonas aeruginosa* [9]. Other gram-positive cocci frequently isolated from bone samples include *Staphylococcus epidermidis* and other coagulase-negative staphylococci, beta-haemolytic streptococci, and diphtheroids. Among the Enterobacterales, *Escherichia coli*, *Klebsiella pneumonia*, and *Proteus* spp. are the most common pathogens. Obligate anaerobes (e.g., *Finegoldia magna*, *Clostridium* spp., or *Bacteroides* spp.) are generally less frequently cultured from bone samples, but this depends on the method by which the bone fragments are taken and transported to the laboratory. A recent prospective study has reported that molecular techniques (16S rRNA sequencing) applied to the assessment of bone biopsies could identify more anaerobes and gram-positive bacilli compared to conventional techniques (86.9% vs. 23.1%, *p* = 0.001, and 78.3% vs. 3.8%, *p* < 0.001, respectively) [10]. According to some recent studies using molecular techniques [10,11], the majority of DFOs are of polymicrobial origin, with a high prevalence of strict anaerobes, reinforcing the idea that the environment of bacteria involved in DFOs for most microorganisms is likely to promote the organization of bacterial communities in biofilms [12].

The poorer outcomes of DFOs compared to ST-DFIs seem to be related to the reduced activity of most antibiotics in the setting of a chronic bone infection. The main reasons are (i) the low diffusion of most antibiotics into the infected bone tissues, especially if chronically infected and necrotic; (ii) the usual empirical antibiotic therapy of DFOs due to the difficulty encountered in organizing bone biopsy, despite the low correlation between the culture results of bone versus non-bone samples reported in numerous studies; and (iii) the very specific environment of the bacteria involved in chronic osteomyelitis.

## 3. Histology

Bone and joint infections (BJIs) complicating a diabetic foot ulcer are the results of the extension to the infection that involves the skin and soft tissues overlying a bony prominence. There are almost no cases where the pathogens reach bone or joints from the bloodstream. While medullar bone or joint may be directly inoculated in the case of puncture wounds, the first osseous structure generally involved during DFO is the periosteum and cortical bone with secondary extension to the bone marrow (i.e., medullar bone). According to Hofmann et al. [13], this kind of “centripetal” infection is defined as osteitis rather than osteomyelitis, although the latter term is rarely used by authors.

Available data on the histological features of DFO are scarce. The usual histomorphological aspects reported in patients with DFO diagnosed on a clinical and imaging basis are varied, including a typical aspect of osteomyelitis, bone necrosis, myelofibrosis, and normal bone as well [14]. Of note, the histomorphology of unaffected foot bone appears mostly normal in diabetic patients with neuropathy and peripheral artery disease [14]. According to Aragon-Sanchez et al., three different histomorphological abnormalities can be found in DFO: acute (destroyed bone, and infiltrations of polymorphonuclear granulocytes at cortical sites and inside the bone marrow usually associated with congestion or thrombosis of medullary or periosteal small vessels); chronic (destroyed bone, and infiltrations of lymphocytes, histiocytes, and/or plasmatic cells at cortical sites and inside the bone marrow); and acute exacerbation of chronic osteomyelitis (a background of chronic osteomyelitis, of which infiltration of polymorphonuclear granulocytes is present) [15]. Areas of fibrosis and medullar oedema can be seen in all cases.

## 4. Biofilm

Beyond the identification of the causative pathogens from bone samples, it is also important to characterize their phenotypic state, which significantly influences the antibacterial effect of most antibiotics. In most cases of DFO, the infection presents as chronic because of the frequent association with the loss of sensation allowing the indolent progression of the infectious process. It is common to oppose planktonic to sessile microorganisms, the former being responsible for acute infections and the latter for chronic infections. Indeed, there are not only two sorts of these microorganisms, but rather a continuum in the metabolic status leading to numerous different types of bacterial cells. Planktonic exhibit a high metabolic and dividing activity exposing them to the activity of most antibiotics. These microorganisms may be involved in rare cases of acute DFO, generally due to highly virulent bacteria with a high tropism for bone tissues, such as *S. aureus*. On the contrary, sessile bacteria do not express many targets to the antibacterial activity of most antibiotics, which is the cause of adaptive resistance and present as “small colony variants” (SCVs) with significant phenotypic alterations in culture [16]. Some of these bacteria can enter a dormant state called “persister cells” [16]. These bacterial cells do not respond to the antibacterial activity of most antibiotic agents and are suspected to be at the origin of latent and recurrent infections. A bacterial phenotype switch to SCVs is facilitated in some specific conditions, such as a biofilm environment, internalization in host cells including osteoblasts, and in the presence of antibiotics [17]. The chronic process of BJIs facilitates the development of biofilms in bone tissues as reported in two independent studies. So far, the presence of biofilm as cause of a difficult to treat infection can only be demonstrated with specific histological studies, electron microscopy, or by mucoid properties of the isolated bacteria; however, these data are often presented in experimental studies but almost never in clinical studies. Baudoux et al. found that about two-thirds of bone samples contained biofilm assessed by both crystal violet staining and electronic microscopy [18]. Eight years later, Johani et al., using electronic microscopy, identified biofilm structures in 80% of the bone specimens [11]. The authors found biofilm structures to be mainly localized at the surface of the periosteum and compact bone while they were absent from compact bone interface. Malone et al. also demonstrated the presence of biofilms within bone samples taken at the bone margins in diabetic patients who underwent bone resections and amputation of the foot [19]. Infected and proximal bone samples from 14 patients who had undergone bone resection or amputation for the treatment of a DFO were examined by electronic microscopy. In half of these cases, microorganisms, including planktonic bacterial cells and aggregates embedded in biofilm structures, were detected in both infected bone samples and the corresponding proximal bone margin [19]. Still, we are missing a simple serum biomarker of biofilm formation in vivo to monitor the presence of a prosthetic infection or an osteomyelitis with biofilm.

Sessile bacterial cells present in biofilm structures are characterized by a higher tolerance toward antimicrobial agents compared to bacteria involved in other types of infection. In addition to their decreased penetration through the biofilm matrix, most antibiotics exhibit diminished antibacterial activity due to a lack of intracellular accumulation and the metabolically inactive status of the bacteria [16]. In biofilms, gene transfer of virulence factors and antibiotic-resistant genes are more likely to transfer between bacteria and alterations in the biofilm environment, especially lower pH and oxygen, which impair the antibacterial activity of some antibiotics [16]. The presence of biofilms produces an intense activation of the immune matrix without effective eradication, which causes collateral damage to surrounding tissues and local chronic inflammation [12]. The production of auto-inflammatory cytokines results in bone damage in addition to bacterial virulent effects.

## 5. Anti-Biofilm Effect of Antibiotics

The term “biofilm” does not refer to a single entity as its structure may significantly differ according to the pathogens involved in its constitution and the level of its maturity. Older (mature) biofilms have more complex structures and are more resistant to antibiotic activity [16]. The cut-off in the age of the biofilms beyond which no antibiotic effect can be expected is unknown, but the clinical experience in prosthetic joint infections suggests a value of around 3–4 weeks [20,21]. It is difficult to separate antibiotics with and without anti-biofilm activity. Some antibiotics, especially those that recently appeared on the market, have been shown to have an anti-biofilm activity; however, for most of them this was only based on in vitro models using young biofilms [22]. Of note, clinical efficacy in biofilm-related infections (e.g., prosthetic joint infections) have been established only for rifampicin and fluoroquinolones (ciprofloxacin and levofloxacin) [23,24,25].

Anti-biofilm antibiotics should (i) achieve concentrations above the minimal inhibitory concentration of the targeted pathogens; (ii) exhibit maintained antimicrobial effects in the biofilm environment characterized by high protein concentrations, a low local oxygen pressure, a reduced pH, and high bacterial inoculum with an intense exchange between bacterial cells of nucleic material, including antimicrobial resistance genes; (iii) maintain their activity against bacteria in the stationary phase with reduced metabolism; and (iv) act on adhering bacteria. A bactericidal effect is likely to be beneficial given the local immunosuppression within the biofilm due to the difficulty of some components of innate immunity to penetrate the biofilm structures [16]. All these requirements may explain the lack of correlation between the bactericidal activity and the anti-biofilm activity in comparison with other types of infection where bacteria are in the exponential growth phase. This is the case for vancomycin, a bactericidal agent with activity against most staphylococcal strains, which has no effect on staphylococcal biofilms assessed in the animal tissue-cage model [26]. Given the ability of some bacteria to invade and survive in a wild range of host cells, intracellular concentration and activity may be of interest to combat biofilm-related bone and joint infections [27]. The intra-osteoblastic bactericidal effect has been reported with fosfomycin, linezolid, tigecycline, oxacillin, rifampicin, ofloxacin, and clindamycin, while ceftaroline and teicoplanin have only a bacteriostatic effect and vancomycin and daptomycin have no significant effect on the intracellular bacterial growth [27]. It is worthwhile to not that daptomycin, the unique representant of the cyclic lipopeptide family, exhibits a maintained activity in vitro against bacteria in the stationary phase, but is inactive in the tissue-cage model [26,28]. Among the new long-acting lipoglycopeptides, a satisfactory in vitro anti-biofilm effect has been reported with both dalbavancin and oritavancin, but without demonstration of any clinical efficiency so far [29]. These data emphasize the important value of animal models which should be completed to confirm any in vitro model suggesting an anti-biofilm effect. Of note, the importance of DFO biofilm is obviously higher in the case of non-surgical treatment than in surgical options, i.e., bone resection including amputations. The current recommendations suggest using longer duration of antibiotic treatments in cases where no chronically infected bone has been removed (i.e., medical treatment of DFO) [30]. On the other hand, there is no strong recommendation to use antibiotics with antibiofilm activity in these settings.

## 6. Antibiotic Therapy

### 6.1. Current Data

While the role of biofilms in the pathogenicity of chronicisation of DFUs has been extensively studied, this is not the case for DFOs. Systemic or topical administrations of antibiotics have not shown efficacy for the management of non-healing DFUs [12]. The reasons are unclear, but the difficulties in obtaining local satisfactory antibiotic concentrations and the large number of bacteria organized in complex pathogroups probably act negatively, in addition to the usual limitations of the antibiotic activity in biofilms [12] (Figure 1).

When examining the efficiency of antibiotic regimens for treating DFO, it is of utmost importance to consider whether the patients were treated with or without resection of the infected bone tissues. The main issue for the medical and surgical management of DFOs is the persistence of chronically infected bone tissues and residual non-necrotic infected bone tissues, respectively. The clinical implication of positive bone margins is a matter of debate, but the International Working Group on the Diabetic Foot recommends obtaining a specimen of bone for culture at the stump of the resected bone [30]. The task for the antibiotic treatment, therefore, depends on the type of infected bone tissues to treat. The presence of biofilm structures in the majority of chronically infected bones in patients diagnosed with DFO raises the question of the benefit of antibiotics with anti-biofilm activity in these settings. Recommendations concerning the choice of antibiotic regimens for the treatment of DFO are mostly based on expert opinions, rather than evidence-based, given the absence of randomized controlled studies that address this question [30]. Some general rules used for the treatment of chronic osteomyelitis can, however, be applied to DFO, including the use of antibiotics with high diffusion into bone (i.e., a bone/blood ratio > 0.3) and good oral bioavailability (i.e., >90%) [31]. Antibiotics that achieve the highest bone-to-serum concentration ratios (i.e., fluoroquinolones, sulfamides, tetracyclines, macrolides, rifampicin, fusidic acid, and oxazolidinones) are also those with the highest oral bioavailability, which makes these agents good candidates for prolonged treatment of outpatients with osteomyelitis (Table 1). One retrospective study suggested that patients treated preferentially with anti-biofilm antibiotic regimens selected based on transcutaneous bone biopsy for the medical treatment of staphylococcal and gram-negative bacilli DFO, respectively, had better remission rates than the other patients treated conventionally (i.e., with standard antibiotics based on superficial samples results like swabs) [32].

The question of the need for using anti-biofilm antibiotics for the treatment following bone resection or amputation for DFO is also a matter of debate. The identification of bacterial cells aggregated in biofilm structures advocates for using anti-biofilm antibiotics once the culture results are available. On the other hand, these bacterial aggregates seem to be associated with planktonic bacterial cells and located on the bone surface, which suggests the immature rather than mature status of the biofilm structures visualized on the corresponding proximal bone margins [19].

No data support any beneficial effect of anti-biofilm antibiotic regimens for possibly reducing the duration of DFO antibiotic treatment. Of note, most of the patients—who were enrolled in a randomized multicenter study of patients with DFO treated medically, suggesting the equivalence of between a 6- and 12-week duration—were given rifampicin and/or fluoroquinolones (either levofloxacin or ciprofloxacin) [33]. Examples of clinical studies that addressed the outcome of patients with DFO treated medically are shown in Table 2. A summary of the potential indications of anti-biofilm antibiotics and the duration of treatment according to the type of diabetic foot infections is presented in Figure 2.

### 6.2. Rifampicin and Fluoroquinolones for the Treatment of DFO

In the end, in clinical studies it is thought that some antibiotic associations are better than others because of their proven anti-biofilm efficacy in experimental studies. No studies have compared the outcome of treatments of DFOs according to the presence or absence of biofilm. Cohort studies suggest a better outcome when rifampicin or fluoroquinolones are used, but no clear causal relationship has been established with the anti-biofilm effect of these molecules, which have other interesting characteristics in the treatment of bone infections. Nothing indicates that their effect is not only additive to the associated antibiotic and that the therapeutic success comes from the use of two molecules rather than one with interesting properties in the treatment of bone infections. Rifampicin is the only antibiotic for which significant activity against staphylococcal biofilms has been established in vitro, in animal models, and in human infections. Rifampicin demonstrated potent activity against persister cells in biofilms that exceed that of any other currently available antibiotic [38]. As such, rifampicin should not be considered as an “adjuvant” therapy, but rather, as the “effector” antibiotic when combined with a companion for the treatment of biofilm-related infections. One of the weaknesses of rifampicin is the risk of the emergence of resistant mutants in cases of high bacterial inoculum, especially when rifampicin is used as a monotherapy. There is no convincing data from clinical studies that may help prioritize the choice of the most appropriate companion to associate with rifampicin. The possible interaction with a companion metabolized by the liver is an important parameter to consider when choosing the rifampicin regimen. Rifampicin is, indeed, an inducer of cytochrome P-450 oxidative enzymes and the P-glycoprotein transport system [39]. Therefore, the serum and tissue concentrations of antibiotics that are substrates of these enzymes are exposed to decreased values. This is the case for clindamycin, linezolid, moxifloxacin, and cotrimoxazole, which are exposed to a reduction of 30–40% of serum concentrations [40,41,42]. Conversely, fluoroquinolones and beta-lactams are not exposed to this risk [43,44,45].

Among the fluoroquinolones, levofloxacin is the most frequently used antibiotic against susceptible strains, in combination with rifampicin, for the treatment of staphylococcal osteomyelitis, with biofilms potentially involved due to low hepatic metabolism, one daily administration, and high tissue concentrations [46,47]. Data from in vitro and clinical studies suggest that fluoroquinolones are for gram-negative bacilli biofilms what rifamycins are for staphylococcal biofilms [20]. Rifampicin combinations should not be debuted before the culture results are available (i.e., as empirical treatment) to avoid rifampicin monotherapy in situations where the staphylococcal strain is susceptible to rifampicin but resistant to the antibiotic associated with rifampicin. Rifampicin combination therapy should be continued until the completion of the treatment for staphylococcal osteomyelitis because the risk of the emergence of resistant mutants is particularly high with these bacteria [48]. For gram-negative bone infections treated with fluoroquinolones, a combination treatment is only required during the first days of treatment, usually with cephalosporins, and then the treatment can be completed with fluoroquinolone monotherapy [49].

The potential benefit of rifampicin combinations in the medical management of patients with DFO remains a matter of debate. Anti-biofilm antibiotic treatments using rifampicin in combination with a fluoroquinolone (ofloxacin) have already been assessed in patients treated medically for DFOs more than 20 years ago [50]. Remission defined as the disappearance of all signs and symptoms of infection at the end of the treatment and the absence of relapse during an average follow up of 22 months was achieved in 76.5% of the cases. More recently, a large U.S. multicenter retrospective study compared the outcomes of patients treated for a DFO with or without rifampicin for 6 weeks [51]. The population included 130 patients treated with rifampicin and 6,044 patients treated without rifampicin. Lower event rates (e.g., amputation after 90 days of diagnosis and deaths) were observed among the rifampicin group (35 of 130 (26.9%) vs. 2,250 of 6,044 (37.2%); *p* = 0.02). The difference between the event rates remained significant after controlling for potentially confounding parameters (odds ratio, 0.65; 95% CI, 0.43–0.96; *p* = 0.04) [51].

### 6.3. Data Expected in the Next Future

Given the retrospective nature of Wilson’s study, a prospective, randomized, double-blind U.S. multicenter study is currently recruiting with the intention to confirm the results of the previous report [52]. In the rifampicin arm, a 6-week rifampicin treatment is “added” to any other antibiotic chosen based on the results of the bone biopsy culture. The primary objective is amputation-free survival in each group. The inclusion criterion is a diagnosis of diabetic foot osteomyelitis, as defined by the International Working Group on the Diabetic Foot rather than on exclusively bone biopsy results. As for Wilson’s study, the benefit of rifampicin combinations in patients treated medically might not be established given the fact that patients included can undergo debridement without any precision as to what extent this included bone resection. In addition, using rifampicin as an “adjuvant” for gram-negative DFOs as planned in this study is not supported by the amount of data currently available in the literature on this subject. Finally, the choice of a unique rifampicin daily dosage of 600 mg may lead to low tissue concentrations in obese patients. The results of this important study are awaited in 2024.

### 6.4. Limitations of Rifampicin-Fluoroquinolones for the Treatment of DFO

Rifampicin or fluoroquinolone treatment may lead to the emergence of antimicrobial resistance due to the selection of pre-existing resistant mutants in the bacterial population present in the infected site. The most critical period is therefore the initiation of the treatment when the bacterial inoculum is still high. It seems prudent to consider the initiation of rifampicin treatment after surgery, if required, since surgery can result in a rapid and drastic reduction in the bacterial load in the infected tissues. In patients with inflammatory signs of the foot without indication for surgery, rifampicin is best started after an initial phase of bactericidal and low risk for selection of antimicrobial resistance antibiotic therapy, such as beta-lactams and/or glyco(lipo) peptides. For the same reasons, the choice of the rifampicin companion should be based on bone culture results given the low correlation between bone and non-bone culture results [4,7]. Routine use of rifampicin combinations is also limited by the frequency of adverse events dominated by nausea, vomiting, hepatic toxicity leading to withdrawing rifampicin (in up to 39% in the Lesens study [8]), and the risk of drug interactions with medications metabolized by the cytochrome P450 system, including warfarin, corticosteroids, thyroid hormones, and some antidiabetic agents. The use of rifabutin, a weaker inducer, with potent biofilm activity in vitro in place of rifampicin may be of clinical interest [53]. It seems prudent to avoid using rifampicin in patients with a suspicion of active tuberculosis due to the risk of the emergence of *Mycobacterium tuberculosis* rifampicin-resistant mutants. The high oral bioavailability of rifampicin and fluoroquinolones allow starting the treatment orally, as suggested in a recent RCT [54]. It is, however, important to remember to take rifampicin on an empty stomach and to avoid concomitant use of aluminum- and magnesium-containing antacids and sucralfate, as well as with other metal cations, such as calcium and iron with fluoroquinolones [55].

## 7. New Anti-Biofilm Modalities

New alternatives to antibiotic strategies against biofilms have emerged, especially bacteriophages, antimicrobial peptides, and nanotechnologies [56,57,58]. While these therapeutic tools have been largely assessed for the treatment of diabetic foot ulcers, the data for DFOs are scarce.

## 8. Conclusions

DFO is a frequent complication associated with diabetic foot infections. Gram-positive cocci, especially *S. aureus* dominate in Western countries while gram-negative, especially *Pseudomonas* spp., are more prevalent in warm-climate countries. The optimal antibiotic regimens to be used in patients presenting with a DFO have not yet been established in well-designed studies. The majority of patients presenting with a DFO have chronic bone and joint infection and most of the infected bone tissues contain biofilm structures. The choice of antibiotics depends on the therapeutic option, e.g., medical versus surgical. The use of antibiotic regimens with anti-biofilm activity, especially rifampicin for staphylococcal infections and fluoroquinolones from gram-negative in patients treated medically, is supported by a solid background issued from in vitro, animal models, and clinical studies; however, the use of these regimens in patients with DFOs has not yet been established, except through a couple of retrospective studies. The VA Intrepid study should provide useful data about the place of rifampicin combinations in these settings.

## Figures and Tables

**Figure 1 antibiotics-12-00317-f001:**
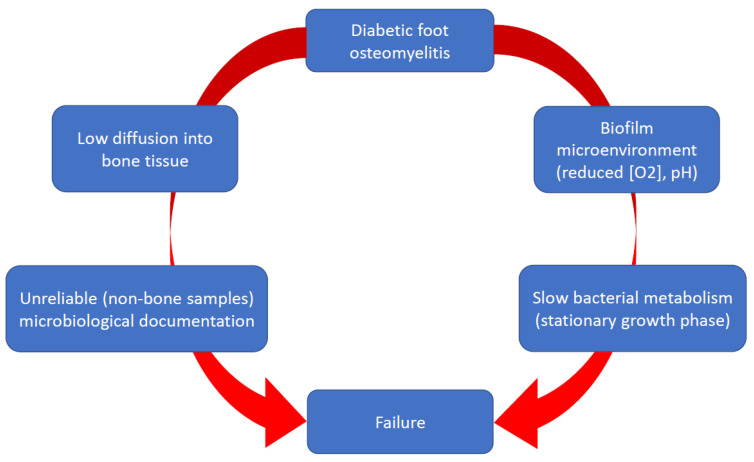
Limitations for the antibiotic therapy of diabetic foot osteomyelitis.

**Figure 2 antibiotics-12-00317-f002:**
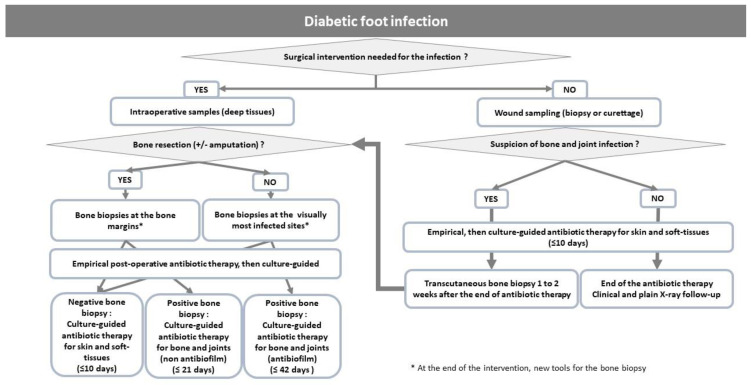
Propositions for the choice of anti-biofilm antibiotics and duration according to the type of diabetic foot infections.

**Table 1 antibiotics-12-00317-t001:** Activity of different antibiotics against biofilm-related infection (modified from Ref. [22]).

Antibiotic	Diffusion into the Biofilm Matrix	Activity against Bacterial Cells in the Stationary Phase	Activity Assessed in Tissue Cage Model and/or Human Study
Fluoroquinolones	Yes	Yes	Yes
Rifampicin	Yes	Yes	Yes
Minocycline	Yes	Yes	No
Fosfomycin	Yes	Yes	No
Daptomycin	Yes	Yes	No
Linezolid	Yes	Reduced	No
Dalbavancin	Reduced	Reduced	No
Vancomycin	Severely reduced	Not known	No
Aminoglycosides	Reduced	Reduced	No

**Table 2 antibiotics-12-00317-t002:** Antibiotic regimens used in patients treated medically for diabetic foot osteomyelitis.

Author (Ref)	N° Patients/Episodes of DFO	Remission (%)	Antibiotic Regimens
Lessens, 2011 [8]	68 medical management (MM): 36 surgical management (SM): 32	MM: 55 SM: 45	amoxicillin-clavulanic acid, ciprofloxacin, cotrimoxazole, rifampicin-fluoroquinolone/clindamycin
Senneville, 2008 [32]	50	64	fluoroquinolone-rifampicin, fluoroquinolone-pristinamycin, fluoroquinolone plus 3rd or 4th generation cephalosporin
Tone, 2015 [33]	40	65	gram-positive cocci: rifampicin plus (levofloxacin, cotrimoxazole, doxycycline, or linezolid) gram-negative bacilli: levofloxacin or ciprofloxacin plus (cefotaxime, ceftriaxone, or cefepime) for the first 2 weeks of treatment, then levofloxacin or ciprofloxacin monotherapy
Embil, 2006 [34]	94/117	80.5	metronidazole, ciprofloxacin, cotrimoxazole, clindamycin, cephalexin amoxicillin +/− clavulanic acid
Valabhji, 2009 [35]	47/53	75	amoxicillin-clavulanic acid, clindamycin-ciprofloxacin, rifampicin-doxycyline
Acharya, 2013 [36]	130	66.9	flucloxacillin-fusidic acid, ciprofloxacin-clindamycin
Lazaro-Martinez, 2014 [37]	46 medical management (MM): 24 surgical management (SM): 22	MM: 79.1 SM: 68.2	amoxicillin-clavulanic acid, ciprofloxacin, cotrimoxazole

## Data Availability

Not applicable.

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
