# Peer review of "Do Anti-Biofilm Antibiotics Have a Place in the Treatment of Diabetic Foot Osteomyelitis?"

_antibiotics, 2023, doi:10.3390/antibiotics12020317_

Round 1

Reviewer 1 Report

No comments

Author Response

We did not find any modifications required by Reviewer 1 except to check the English language

As we informed the Editor, the delay to provide the revised version did not allow us to ask a native-speaker to improve the document regarding the English and we apolize for this

Reviewer 2 Report

The current review is on a topic of relevance and general interest to the readers of the journal. Written for experts on diabetic foot I found the paper to be overall well synthetized with the different parts correctly identified and it summarizes the most discussed questions around the biofilm in diabetic foot osteomyelitis. It is nice to read and I think these works can be useful to encourage researchers to follow the research in the field of diabetic foot.

I have minor comments below.

In the section 5. Anti-biofilm effect of antibiotics. It is missed a brief discussion about the different importance of DFO biofilm in conservative treatment (no surgery) versus in the other completely different scenario with DFO amputation. Please, explain in some added lines about the exigence of antibiotic duration and the importance of biofilm antibiotic in each case.

In Section 6. Antibiotic therapy. When the authors refer to “When examining the efficiency of antibiotic regimens for treating DFO, it is of utmost importance to consider whether the patients were treated with or without resection of the infected bone tissues”. Related to this information it is missed some consideration about the importance of bone proximal positive or negative margins in patients with bone resection regarding the discussion about the distal or proximal positive/negative margins, and its implications in clinical practice.

In the section Rifampicin and fluoroquinolones for the treatment of DFO. When the authors write For gram-negative bone infections treated with fluoroquinolones, a combination treatment is only required during the first days of treatment, usually with cephalosporins, then the treatment can be completed with fluoroquinolone monotherapy” Please provide some evidence (reference) about this affirmation.

When authors explain the results of Wilson’s study (reference 51), could be please summarize only the most important findings? The more infectious disease specialist consultations are not necessary to document it.

Figure 1 is difficult to read. It should be improved for the final document.

Author Response

In the section 5. Anti-biofilm effect of antibiotics. It is missed a brief discussion about the different importance of DFO biofilm in conservative treatment (no surgery) versus in the other completely different scenario with DFO amputation. Please, explain in some added lines about the exigence of antibiotic duration and the importance of biofilm antibiotic in each case.

Authors’ response

We added at the end of the section :

Of note, the importance of DFO biofilm is obviously higher in the case of non-surgical treatment than in surgical options ie., bone resection including amputations. The current recommendations suggest using longer duration of antibiotic treatments in cases where no chronically infected bone has been removed (i.e., medical treatment of DFO) [30]. On the other hand, there is no strong recommendation to use antibiotics with antibiofilm activity in these settings.

In Section 6. Antibiotic therapy. When the authors refer to “When examining the efficiency of antibiotic regimens for treating DFO, it is of utmost importance to consider whether the patients were treated with or without resection of the infected bone tissues”. Related to this information it is missed some consideration about the importance of bone proximal positive or negative margins in patients with bone resection regarding the discussion about the distal or proximal positive/negative margins, and its implications in clinical practice.

Authors’ response

We added at the end of the section :

The clinical implication of positive bone margins is a matter of debate but the International Working Group on the Diabetic Foot recommends obtaining a specimen of bone for culture at the stump of the resected bone [30].

In the section Rifampicin and fluoroquinolones for the treatment of DFO. When the authors write “For gram-negative bone infections treated with fluoroquinolones, a combination treatment is only required during the first days of treatment, usually with cephalosporins, then the treatment can be completed with fluoroquinolone monotherapy” Please provide some evidence (reference) about this affirmation.

Authors’ response

We added the following reference:

Legout, L.; Senneville, E.; Stern, R.; Yazdanpanah, Y.; Savage, C.; Roussel-Delvalez, M.; Rosele, B.; Migaud, H.; Mouton, Y. Treatment of bone and joint infections caused by Gram-negative bacilli with a cefepime-fluoroquinolone combination. Clin. Microbiol. Inf.. 2006, 12(10), 1030–1033.

When authors explain the results of Wilson’s study (reference 51), could be please summarize only the most important findings? The more infectious disease specialist consultations are not necessary to document it.

Authors’ response

We agree and we reduced the paragraph about the Wilson’s study in the “current data” section

Figure 1 is difficult to read. It should be improved for the final document.

Authors’ response

We do not understand if the comment is about the quality of the figure or its content

Reviewer 3 Report

The section on microbiology interactions with diabetic foot, I suggest including a graphical representation for better self-explanatory. 

Add molecular mechanism methodology explaining the cellular interactions of biofilm effect on antibiotic resistance and human cellular pathophysiology. 

Add future prospects of the new modalities. 

Author Response

The section on microbiology interactions with diabetic foot, I suggest including a graphical representation for better self-explanatory. 

Authors’ s response:

We added a figure (Figure 1) at the end of chapter 2

Add molecular mechanism methodology explaining the cellular interactions of biofilm effect on antibiotic resistance and human cellular pathophysiology. 

Authors’ response :

We added the following paragraph at the end of Chapter 4

Sessile bacterial cells present in biofilm structures are characterized by a higher tolerance towards antimicrobial agents compared to bacteria involved in other types of infection. In addition to their decreased penetration through the biofilm matrix, most antibiotics exhibit diminished antibacterial activity due to a lack of intracellular accumulation and the metabolically inactive status of the bacteria [16]. In biofilms, gene transfer of virulence factors and antibiotic-resistant genes are more likely to transfer between bacteria and alterations in the biofilm environment especially lower pH and oxygen impair the antibacterial activity of some antibiotics [16]. The presence of biofilms produces an intense activation of the immune matrix without effective eradication which causes collateral damage to surrounding tissues and local chronic inflammation [12]. The production of auto-inflammatory cytokines results in bone damage in addition to bacterial virulent effects.

Add future prospects of the new modalities. 

Authors’ response:

We added a  new chapter (#7) entitled “ New anti-biofilm modalities” :

New alternatives to antibiotic strategies against biofilms have emerged especially bacteriophages, antimicrobial peptides and nanotechnologies [57-60]. While these therapeutic tools have been largely assessed for the treatment of diabetic foot ulcers, the data for DFOs are scarce.

Reviewer 4 Report

This paper is a review presenting the available data on antibiotic therapy of diabetic foot osteomyelitis with a focus on possible anti-biofilm effects. However, it is not a metanalysis but a synthetic summary of the available studies in diabetic foot infection where it may be present a biofilm in case of osteomyelitis.

So far, the presence of biofilm as cause of a difficult to treat infection can be demonstrated only with specific histological studies or by electron microscopy or by mucoid properties of the isolated bacteria; these data are often presented in experimental studies but almost never in clinical studies. At the end, in clinical studies, it is thought that some antibiotic associations are better than others because of their proven anti-biofilm efficacy in experimental studies. Still, we are missing a simple serum biomarker of biofilm formation in-vivo to monitor the presence of a prosthetic infection or an osteomyelitis with biofilm.

These are limitations which should be addressed for accepting the paper for publication.

At page 4 it is used the term “cyclines” instead of the more adequate “tetracycline”; indeed, the “cyclines” are even the proteins controlling the cell-cycle.

Author Response

This paper is a review presenting the available data on antibiotic therapy of diabetic foot osteomyelitis with a focus on possible anti-biofilm effects. However, it is not a metanalysis but a synthetic summary of the available studies in diabetic foot infection where it may be present a biofilm in case of osteomyelitis.

So far, the presence of biofilm as cause of a difficult to treat infection can be demonstrated only with specific histological studies or by electron microscopy or by mucoid properties of the isolated bacteria; these data are often presented in experimental studies but almost never in clinical studies. At the end, in clinical studies, it is thought that some antibiotic associations are better than others because of their proven anti-biofilm efficacy in experimental studies. Still, we are missing a simple serum biomarker of biofilm formation in-vivo to monitor the presence of a prosthetic infection or an osteomyelitis with biofilm.

These are limitations which should be addressed for accepting the paper for publication.

We included the three sentences proposed by Reviewers #4 without modifying them in the corresponding paragraphs of the paper

At page 4 it is used the term “cyclines” instead of the more adequate “tetracycline”; indeed, the “cyclines” are even the proteins controlling the cell-cycle.

Authors’ response:

We edited the corresponding text accordingly